# FROM PATCHES TO GRAPHS:
# TOWARDS IMAGE DIFFUSION MODELS WITH GNNS

## ABSTRACT

Diffusion models have achieved remarkable success in high-quality image generation, typically using convolutional neural networks (CNNs) or Vision Transformers (ViTs) as backbone architectures. However, CNNs may struggle with capturing long-range dependencies, while ViTs can be computationally intensive due to their attention mechanisms. We propose the Diffusion Image GNN (DiG), a novel architecture that leverages graph-based modeling within diffusion models. By representing image patches as nodes in a graph and connecting them based on spatial relationships, DiG efficiently captures both local and global dependencies and naturally handles multi-scale features. Empirical results demonstrate that DiG achieves competitive Frechet Inception Distance (FID) scores compared to state-of-the-art methods. To our knowledge, this is the first application of graph neural networks as a backbone within diffusion models for image generation, opening new avenues for research in generative modeling.

## 1 INTRODUCTION

Diffusion models (Sohl-Dickstein et al., 2015; Song & Ermon, 2019; Ho et al., 2020; Nichol & Dhariwal, 2021; Croitoru et al., 2023) have emerged as a powerful class of generative models, achieving state-of-the-art performance in high-quality image generation (Dhariwal & Nichol, 2021; Rombach et al., 2022). They have been successfully applied to a diverse range of tasks, including image generation (Choi et al., 2021; Saharia et al., 2022b; Rombach et al., 2022), text-to-image synthesis (Jiang et al., 2022; Ramesh et al., 2022), video generation (Ho et al., 2022a;b; Blattmann et al., 2023; Xing et al., 2023), among others (Jing et al., 2022a; Wolleb et al., 2022).

Alongside advancements in the mathematical framework of diffusion models (Song et al., 2020; 2021b; Watson et al., 2022; Bao et al., 2022; Dockhorn et al., 2021; Kingma et al., 2021; Song et al., 2021a; Vahdat et al., 2021; Lu et al., 2022), the choice of backbone architectures plays a pivotal role in determining their overall performance. Historically, Convolutional Neural Networks (CNNs) (LeCun et al., 1998; He et al., 2016), such as U-Net (Ronneberger et al., 2015), have served as the de facto standard architecture in modern computer vision systems. However, recent developments have seen the introduction of Vision Transformers (ViT) (Dosovitskiy et al., 2021), which utilize attention mechanisms for visual tasks. Building on this architecture, new models (Hatamizadeh et al., 2023; Peebles & Xie, 2023; Bao et al., 2023) have emerged that adapt the transformer-based design (Vaswani, 2017) for diffusion models, demonstrating competitive performance in generative tasks. Following these advances, State Space Models (SSM) (Gu et al., 2021) have shown strong capabilities in handling long-sequences, contributing to a new class of diffusion models that capture fine-grained representations of images (Yan et al., 2024). These backbone architectures process input data in fundamentally different ways. CNNs operate at the pixel level, applying convolutional kernels across a grid of pixels to capture local features. In contrast, Transformers and SSMs process images by treating patches as sequential tokens, where each patch represents a portion of the image. By computing attention across this sequence of patches, Transformers fully connect them, allowing the model to capture global dependencies across the image.

An alternative approach involves representing images as compositions of their parts within a flexible grid structure. Graph-based models (Kipf & Welling, 2016; Wu et al., 2020) are particularly well-suited for this representation, as they naturally facilitate the modeling of complex objects by capturing the relationships between different parts. Unlike Transformers, which connect all patches

through attention mechanisms – creating a fully connected graph – graph-based models can naturally handle grid-like data and connect different scales of the image. By treating image components as nodes and establishing edges between them, graphs allow for efficient computation of relationships and can capture complex patterns inherent in images.

In this work, we propose a novel backbone architecture called the *Diffusion Image GNN (DiG)*, which leverages graph-based modeling within image diffusion models. DiG processes images by treating a patch as a node, and it constructs a structured graph by leveraging the spatial grid to capture spatial locality of node features, and connecting multi-scale patch nodes to efficiently propagate global information. Our contributions are summarized as follows:

- We introduce DiG, a novel graph-based network backbone for diffusion models that processes images as graphs, capturing both local and global dependencies. We further propose two strategies to handle multi-scale features, which connects different levels of granularity within the graph nodes representing image patches.

- We provide empirical evidence showing that DiG achieves competitive performance in FID scores compared to popular Transformer-based approaches, highlighting the effectiveness of graph-based representations in diffusion models.

To the best of our knowledge, this is the first work to apply graph neural networks (GNNs) as a backbone for image diffusion models with competitive performance, and we hope our work will inspire the community to further explore GNN-based backbones.

## 2 BACKGROUND & RELATED WORK

### 2.1 DIFFUSION MODELS

Diffusion models (Ho et al., 2020; Sohl-Dickstein et al., 2015; Song et al., 2021b) are a class of generative models that synthesize new data by progressively adding noise to existing data and then reversing this process to recover samples from the original data distribution. This approach involves two key stages: a forward noising process and a reverse denoising process.

In the forward process, an original data point $\boldsymbol{x}_0$ is gradually corrupted by adding noise at each time step, resulting in a sequence of increasingly noisy data points $\boldsymbol{x}_1, \boldsymbol{x}_2, \ldots, \boldsymbol{x}_T$. This process is modeled as a Markov chain, where each state depends only on the immediate previous state. The forward process is defined by the joint distribution $q(\boldsymbol{x}_{1:T} \mid \boldsymbol{x}_0) = \prod_{t=1}^{T} q(\boldsymbol{x}_t \mid \boldsymbol{x}_{t-1})$, where $q(\boldsymbol{x}_t \mid \boldsymbol{x}_{t-1})$ represents the transition probability at each time step $t$. At each step, Gaussian noise is added according to $q(\boldsymbol{x}_t \mid \boldsymbol{x}_{t-1}) = \mathcal{N}\left(\boldsymbol{x}_t; \sqrt{\alpha_t}\,\boldsymbol{x}_{t-1}, \beta_t \mathbf{I}\right)$, with $\alpha_t$ and $\beta_t$ being parameters that control the noise schedule and satisfying $\alpha_t + \beta_t = 1$.

In the reverse process, the goal is to invert the forward diffusion by iteratively denoising $\boldsymbol{x}_T$ back to $\boldsymbol{x}_0$. This reverse process is also modeled as a Markov chain with learned Gaussian transitions $p_\theta(\boldsymbol{x}_{t-1} \mid \boldsymbol{x}_t) = \mathcal{N}\left(\boldsymbol{x}_{t-1}; \mu_\theta(\boldsymbol{x}_t, t), \Sigma_\theta(\boldsymbol{x}_t, t)\right)$, where $\mu_\theta(\boldsymbol{x}_t, t)$ and $\Sigma_\theta(\boldsymbol{x}_t, t)$ are the mean and covariance predicted by a neural network parameterized by $\theta$. Rather than directly learning $\mu_\theta(\boldsymbol{x}_t, t)$ (Bao et al., 2022), diffusion models often reformulate the problem as a noise prediction task. By leveraging the property that any $\boldsymbol{x}_t$ can be expressed as a function of $\boldsymbol{x}_0$ and added noise $\epsilon_t$ as $\boldsymbol{x}_t = \sqrt{\bar{\alpha}_t}\,\boldsymbol{x}_0 + \sqrt{1 - \bar{\alpha}_t}\,\epsilon_t$, where $\bar{\alpha}_t = \prod_{s=1}^{t} \alpha_s$ and $\epsilon_t \sim \mathcal{N}(0, \mathbf{I})$, the model can be trained to predict $\epsilon_t$ directly. The training objective then becomes minimizing the mean squared error between the predicted noise $\epsilon_\theta(\boldsymbol{x}_t, t)$ and the true noise $\epsilon_t$, formulated as $\mathcal{L}_{\text{simple}}(\theta) = \mathbb{E}_{t, \boldsymbol{x}_0, \epsilon_t}\left[\|\epsilon_t - \epsilon_\theta(\boldsymbol{x}_t, t)\|_2^2\right]$. This loss function is feasible with a fixed covariance (Nichol & Dhariwal, 2021) and is evaluated by sampling a random time step $t$, generating $\boldsymbol{x}_t$ using the known $\boldsymbol{x}_0$ and $\epsilon_t$, and training the network to predict $\epsilon_t$.

Diffusion models can be extended to conditional generation tasks (Dhariwal & Nichol, 2021), where the objective is to generate data conditioned on additional information $c$, such as class labels. In this scenario, the noise prediction network incorporates the conditioning information, modifying the training objective to $\mathcal{L}_{\text{cond}}(\theta) = \mathbb{E}_{t, \boldsymbol{x}_0, c, \epsilon_t}\left[\|\epsilon_t - \epsilon_\theta(\boldsymbol{x}_t, t, c)\|_2^2\right]$.

## 2.2 ARCHITECTURES FOR IMAGE DIFFUSION MODELS

Convolutional Neural Networks (CNNs) (LeCun et al., 1998; He et al., 2016) have long been the cornerstone of computer vision, applied to a wide array of visual tasks (Krizhevsky et al., 2012; Ren et al., 2016; Long et al., 2015). Specifically, for image generation, U-Net (Ronneberger et al., 2015) has been extensively used in diffusion models (Ho et al., 2020; Dhariwal & Nichol, 2021; Nichol & Dhariwal, 2021; Ramesh et al., 2022; Rombach et al., 2022; Saharia et al., 2022a). U-Net features a symmetric encoder-decoder architecture designed for pixel-wise prediction. It utilizes skip connections (He et al., 2016) to transfer feature maps from the encoder to the corresponding decoder layers, preserving spatial information that might be lost during down-sampling.

Recently, Transformers based on Attention (Vaswani, 2017) have begun to supplant domain-specific architectures due to their scalability and ability to model long-range dependencies (Kaplan et al., 2020; Henighan et al., 2020). This shift has led to the introduction of Vision Transformers (ViT) (Dosovitskiy et al., 2021) as a common architecture for visual tasks (Chen et al., 2021; Lee et al., 2021; Strudel et al., 2021). In this regard, diffusion models built purely with Transformers have shown remarkable performance in image generation. For instance, U-ViT (Bao et al., 2023) adopts a similar architecture to U-Net but replaces CNN layers with Attention layers. DiT (Peebles & Xie, 2023) follows a comparable approach but incorporates techniques from ResNets (He et al., 2016; Goyal et al., 2019) by zero-initializing a selected set of parameters and introducing dimension-wise scaling factors. DiffiT (Hatamizadeh et al., 2023) introduces a time-dependent self-attention mechanism to jointly learn spatial and temporal dependencies. Other works enhance U-Net by integrating self-attention in the low-resolution blocks and building a hybrid architecture, combining CNNs and Transformers (Ho et al., 2020; Nichol & Dhariwal, 2021; Hoogeboom et al., 2023).

DiffuSSM (Yan et al., 2024) attempts to replace attention mechanisms with a more scalable state space model backbone (Gu et al., 2021), generating high-quality images while being FLOP-efficient. This approach opens avenues for applications that require modeling long-range dependencies without the computational overhead associated with attention mechanisms.

## 2.3 PATCHIFICATION FOR IMAGE PROCESSING

Patchification is the key pre-processing step in image diffusion transformers (Bao et al., 2023; Peebles & Xie, 2023; Hatamizadeh et al., 2023) and DiffuSSM (Yan et al., 2024), which transforms an image input to a set of tokens. Here, a raw input stands for a noisy image $\mathbf{x}_t \in \mathbb{R}^{h \times w \times c'}$ at time step $t$, where $h$ and $w$ denote the height and width with $c'$ channels. Then patchification involves partitioning the image into non-overlapping patches of size $s \times s$, resulting in $n = \frac{h \times w}{s^2}$ patches. Each patch is linearly embedded to form a new representation of $d$ embedding dimension. The granularity of the patch size $s$ significantly impacts quality and computational efficiency due to the quadratic complexity of attention mechanisms (Bao et al., 2023). The attention mechanism computes relationships between all pairs of tokens, effectively modeling the input as a fully connected graph. While this captures global dependencies, it leads to substantial computational overhead. Towards mitigating this, FlexAttention (Li et al., 2024) conducts patchification at both lower and higher resolutions, effectively reduces the number of active tokens. Such a method has yet to be tested on image diffusion models to the best of our knowledge, leaving rooms for future research.

## 2.4 GRAPH-BASED MODELS FOR IMAGES

The applications of Graph Convolutional Networks (GCNs) (Henaff et al., 2015; Kipf & Welling, 2016; Defferrard et al., 2016) in computer vision (Jing et al., 2022b) mainly include point cloud classification (Landrieu & Simonovsky, 2018), scene graph generation (Xu et al., 2017), and action recognition (Wang et al., 2019). These applications leverage naturally constructed graphs to model relationships inherent in the data. For more general applications, the Vision GNN (ViG) (Han et al., 2022) processes image data directly by representing images as graphs, where nodes correspond to image patches, and edges are constructed by connecting nodes with similar visual tokens using k-nearest neighbors.

However, in the context of diffusion models for image generation, there has not yet been a backbone architecture that leverages graph-based modeling. Note that ViG is not directly applicable in diffusion model setting for two main reasons. First, noise injection in the forward process can change the

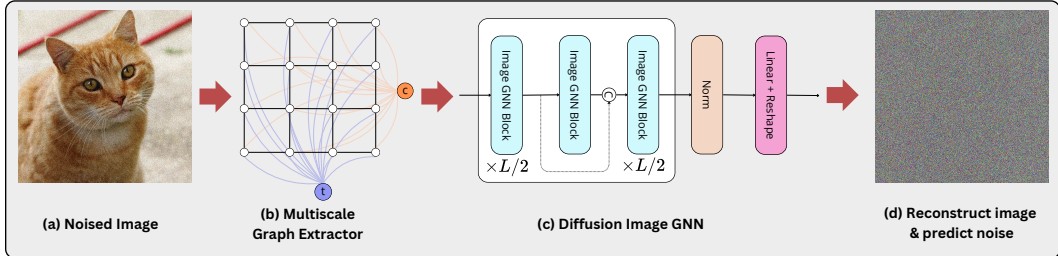

(a) Noised Image     (b) Multiscale Graph Extractor     (c) Diffusion Image GNN     (d) Reconstruct image & predict noise

Figure 1: The process of multiscale graph-based denoising: (a) **Noised Image:** The original image is corrupted by noise $\epsilon_t$; (b) **Multiscale Graph Extractor:** The image is represented as a multiscale graph where each node corresponds to a patch of the image, and two special nodes — time $t$ and context $c$ — are fully connected to the entire grid.; (c) **Image GNN:** DiG blocks are applied to the multiscale graph, capturing features at different scales; (d) **Reconstruction and Noise Prediction:** The processed graph is used to reconstruct the original image and predict the noise added.

results of the nearest-neighbor, leading to a graph connectivity pattern that can change significantly at different time step $t$. Second, at the beginning of the reverse process, which starts from drawing random noise pixels, the extracted patch tokens are essentially i.i.d., meaning the constructed graph does not reflect meaningful visual structure. These challenges motivates our exploration of improved graph representation learning for image generation tasks within diffusion models.

## 3 DIFFUSION IMAGE GNN (DiG)

We propose a novel network backbone architecture for diffusion models, Diffusion Image GNN (DiG), that combines the strengths of graph-based models with diffusion processes, without relying on Transformers. It operates on a multi-scale grid of patches, enabling the model to capture both local and global dependencies. By incorporating patches of different sizes, the model can focus on localized context by primarily attending to neighboring patches. At the same time, it integrates broader spatial information from larger regions, enabling the capture of more complex patterns across the image. This hierarchical patching strategy enhances performance by balancing detailed local features with global spatial understanding, making it particularly effective for image generation tasks in diffusion models.

DiG is summarized in Figure 1. The process begins with a noised image $\boldsymbol{x}_t$ corrupted by noise $\epsilon_t$ (Fig. 1a). The noised image $\boldsymbol{x}_t$ is then represented as a grid of interconnected patches, with time $t$ and context $c$ treated as special nodes fully connected to the grid (Fig. 1b). Scales of an image are constructed using varying patch sizes, represented as an ordered set $\mathcal{P} = \{s_1 < s_2 < \cdots < s_N\}$, where the image size remains fixed, but the graph size changes depending on the selected patch size $s_i$. We introduce two new architectures of DiG, both operating on a grid of connected patches: (1) DiG$_{\text{parallel}}$, which applies a distinct GNN to each scale of the image to capture unique granularities, and (2) DiG$_{\text{mix}}$, which integrates all scales into a unified multiscale graph processed by a single GNN with hierarchical connections between different patch sizes. Both types of DiG facilitate efficient feature learning across multiple resolutions, by using GNN layers for processing graph-based image patches, LayerNorm (Ba et al., 2016; Fey & Lenssen, 2019) for normalization, MLPs for feature transformation, and skip connections to retain original feature information and improve gradient flow (Fig. 1c). The network is optimized to predict the noise $\epsilon_t$ added to the image (Fig. 1d) using objective $\mathcal{L}_{\text{simple}}(\theta)$ or $\mathcal{L}_{\text{cond}}(\theta)$, where $\theta$ collects all trainable parameters in DiG.

Within a DiG block, each GNN layer is designed to handle feature aggregation and updates using the aggregation function $\phi$ and the update function $\gamma$, both parameterized by learnable weights $W_{agg}$ and $W_{update}$, respectively. For a given graph $G = (\mathbf{A}, \mathbf{H}^{(k-1)})$, where $\mathbf{A}$ represents the adjacency matrix and $\mathbf{H}^{(k-1)}$ the node features, each GNN processes feature propagation and transformation across the graph over $K$ hops.

$$\mathbf{H}^{(k)} = \text{GNN}^{(k-1)}(\mathbf{A}, \mathbf{H}^{(k-1)}) = \gamma^{(k)}\left(\phi^{(k)}\left(\mathbf{A}, \mathbf{H}^{(k-1)}, W_{agg}\right), W_{update}\right), \ k = 1, ..., K. \quad (1)$$

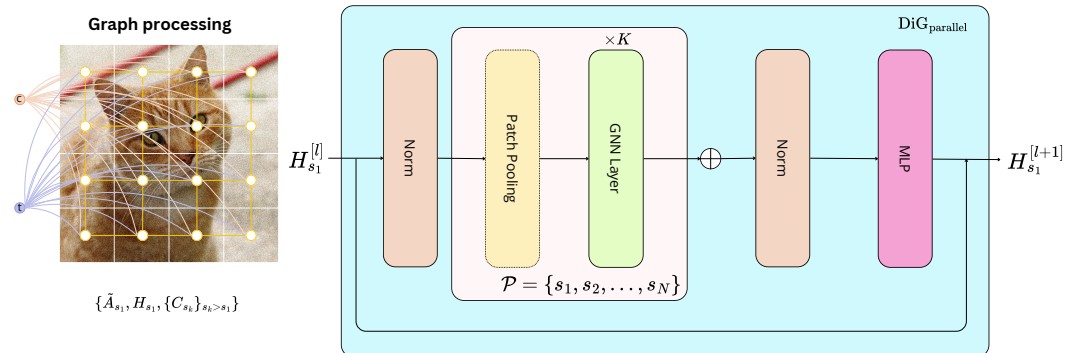

Figure 2: Overview of the **(left)** graph construction, where an image is interpreted as a grid-structured graph, and **(right)** the DiG$_{\text{parallel}}$ block, used to capture different image resolutions and update node features progressively.

To incorporate positional information into the node features, we add a 1-dimensional learnable positional embedding $\mathbf{e}$, following the design of transformers (Vaswani, 2017; Dosovitskiy et al., 2021). Each initial node feature is updated as $\mathbf{H}^{(0)} \leftarrow \mathbf{H}^{(0)} + \mathbf{e}$. This ensures that the positional information of each node is considered during processing.

We now introduce the two types of DiG blocks (shown as cyan blocks in Fig. 1c): DiG$_{\text{parallel}}$ and DiG$_{\text{mix}}$. Both methods share a common *pre-processing* stage that transforms the input image into a graph representation, involving the construction of an initial adjacency matrix $\mathbf{A}$ and subsequent augmentation by adding time and context nodes.

First, node features are extracted from the noised image patches $\mathbf{x}_t$ through a process of patchification. The image is divided into patches, where each patch is treated as a node in a grid-structured graph. For both DiG$_{\text{parallel}}$ and DiG$_{\text{mix}}$, the adjacency matrix $\mathbf{A}_{s_i} \in \mathbb{R}^{n_i \times n_i}$ is constructed at each corresponding patch scale $s_i$, where $n_i = \frac{h \times w}{s_i^2}$. This matrix $\mathbf{A}_{s_i}$ encodes the connections between patches in a grid structure for each resolution $s_i$.

Next, we introduce two special nodes: one for time $t$ and one for context $c$. The time node is encoded using a sinusoidal timestep embedding, while the context node (if provided) is projected using a linear embedding. These special nodes are fully connected to the grid of patch nodes whose adjacency matrix is $\mathbf{A}$. This augmentation results in an augmented adjacency matrix $\tilde{\mathbf{A}} \in \mathbb{R}^{\tilde{n} \times \tilde{n}}$, detailed below and in Appendix A for the full block matrix description. The augmented graph now has $\tilde{n}$ nodes. The corresponding node features are represented as $\mathbf{H} \in \mathbb{R}^{\tilde{n} \times d}$, where $d$ is the feature dimension. This setup ensures that both the *parallel* and *mix* methods can process the entire graph structure, including patch nodes, time, and context nodes, in subsequent steps.

## 3.1 DiG$_{\text{PARALLEL}}$: INDEPENDENT GRAPH PROCESSING ACROSS IMAGE SCALES

DiG$_{\text{parallel}}$ uses a series of GNNs to independently process the image at different scales, capturing unique features at each scale. The node representations at layer $l$ are computed as (see Figure 2):

$$\mathbf{H}_{s_1}^{[l+1]} = \text{DiG}_{\text{parallel}}^{[l]}(\tilde{\mathbf{A}}_{s_1}, \mathbf{H}_{s_1}^{[l]}, \{\boldsymbol{C}_{s_k}\}_{s_k > s_1}). \tag{2}$$

The augmented adjacency matrix $\tilde{\mathbf{A}}_{s_1}$ is formed by adding the time and context nodes to the grid represented by the original adjacency matrix $\mathbf{A}_{s_1}$, and these two nodes are fully connected to the grid. Therefore $\tilde{\mathbf{A}}_{s_1} \in \mathbb{R}^{\tilde{n}_1 \times \tilde{n}_1}$, where $\tilde{n}_1 = n_1 + 2$ when training conditional generative models, else $\tilde{n}_1 = n_1 + 1$. This matrix encodes the relationships between the patch nodes and the special nodes (see Figure 2 left). See Appendix A.1 for the block matrix structure.

To create a coarser graph representation, a pooling mask matrix $\boldsymbol{C}_{s_i} \in \mathbb{R}^{\tilde{n}_i \times \tilde{n}_1}$ is constructed to transition from the finest patch size $s_1$ to a larger patch size $s_i$. This pooling operation downsamples the number of nodes, resulting in a coarser graph that captures larger-scale features of the image (see Figure 2 right). The pooling operator $\text{POOL}(\tilde{\mathbf{A}}_{s_1}, \tilde{\mathbf{H}}_{s_1}, \mathbf{C}_{s_i})$, where $\tilde{\mathbf{H}}_{s_1} = \text{Norm}(\mathbf{H}_{s_1}^{[l]})$, returns

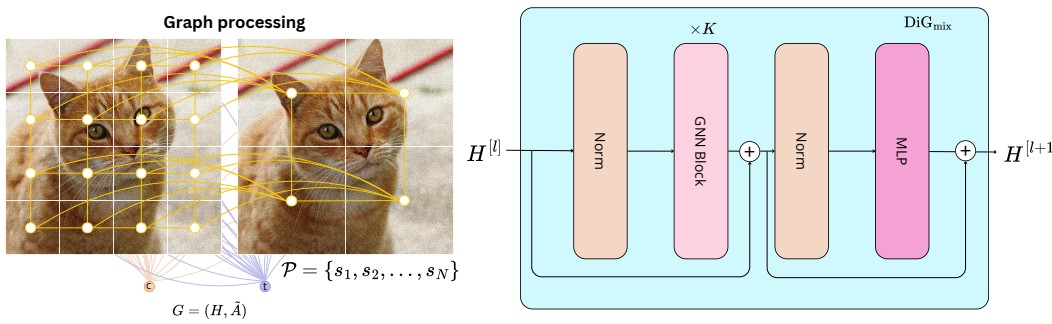

Figure 3: The DiG$_{\text{mix}}$ block takes as input a graph $\mathbf{G}$ representing grid patches at various scales from $\mathcal{P}$ **(left)**. The block processes the graph to produce node features at multiple resolution scales within $\mathcal{P}$ **(right)**.

the coarser node features $\tilde{\mathbf{H}}_{s_i} \in \mathbb{R}^{\tilde{n}_i \times d}$ and adjacency matrix $\tilde{\mathbf{A}}_{s_i} \in \mathbb{R}^{\tilde{n}_i \times \tilde{n}_i}$, representing the image at a larger scale for $s_i > s_1$:

$$(\tilde{\mathbf{H}}_{s_i}, \tilde{\mathbf{A}}_{s_i}) = \text{POOL}(\tilde{\mathbf{A}}_{s_1}, \tilde{\mathbf{H}}_{s_1}, \mathbf{C}_{s_i}) : \quad \tilde{\mathbf{H}}_{s_i} = \mathbf{C}_{s_i}\tilde{\mathbf{H}}_{s_1}, \quad \tilde{\mathbf{A}}_{s_i} = \mathbf{C}_{s_i}\tilde{\mathbf{A}}_{s_1}\mathbf{C}_{s_i}^{\top}. \tag{3}$$

Using the pooling masks $\{\mathbf{C}_{s_i}\}_{s_i > s_1}$, multiple graphs $\mathbf{G}_{s_i} = (\tilde{\mathbf{H}}_{s_i}, \tilde{\mathbf{A}}_{s_i})$ are constructed where each graph corresponds to the image at a different scale, with $\mathbf{G}_{s_1}$ representing the patch image at the lowest scale. Each graph is processed by a GNN layer for up to $K$ hops to update the node features (also see Eq. 1):

$$\tilde{\mathbf{H}}_{s_i}^{(k)} = \text{GNN}_{s_i}^{(k-1)}(\tilde{\mathbf{H}}_{s_i}^{(k-1)}, \tilde{\mathbf{A}}_{s_i}), \quad k = 1, ..., K, \quad \tilde{\mathbf{H}}_{s_i}^{(0)} = \tilde{\mathbf{H}}_{s_i}. \tag{4}$$

Node features from different scales are combined using $\bigoplus$, representing a concatenation operation:

$$\hat{\mathbf{H}} = \text{MLP}\left(\left[\tilde{\mathbf{H}}_{s_1}^{(K)}, \mathbf{H}_{s_2}^{\uparrow}, \ldots, \mathbf{H}_{s_N}^{\uparrow}\right]\right) \tag{5}$$

where $\mathbf{H}_{s_i}^{\uparrow} = \mathbf{C}_{s_i}^{\top}\tilde{\mathbf{H}}_{s_i}^{(K)}$, $\quad \mathbf{H}_{s_i}^{\uparrow} \in \mathbb{R}^{\tilde{n}_1 \times d}$ for $s_i > s_1$ is used to upsample the node features from scale $s_i$ to the finest scale $s_1$. The resulting concatenated feature matrix $\hat{\mathbf{H}} \in \mathbb{R}^{\tilde{n}_1 \times d}$ is then obtained by using an MLP to update the node features, integrating information from multiple resolutions into a unified feature representation. After combining the features, a normalization step using Layer-Norm and an additional MLP are applied, with skip connections added between layers to retain the original feature information , yielding $\mathbf{H}_{s_1}^{[l+1]}$:

$$\mathbf{H}_{s_1}^{[l+1]} = \mathbf{H}_{s_1}^{[l]} + \text{MLP}(\text{Norm}(\hat{\mathbf{H}})). \tag{6}$$

This results in a unified multiscale graph representation, enabling the stacking of multiple DiG$_{\text{parallel}}$ blocks to progressively update the node representation at each layer $l$.

## 3.2 DiG$_{\text{MIX}}$ : Integrated Graph Processing Across Multiple Resolutions

DiG$_{\text{mix}}$, visualized in Figure 3, uses a single GNN module to efficiently learn from a graph that is built using multiple resolutions of the image. At each layer $l$, the node features are updated as:

$$\mathbf{H}^{[l+1]} = \text{DiG}_{\text{mix}}^{[l]}(\mathbf{H}^{[l]}, \tilde{\mathbf{A}}). \tag{7}$$

The unified graph $\mathbf{G} = (\mathbf{H}, \tilde{\mathbf{A}})$, with $\tilde{\mathbf{A}} \in \mathbb{R}^{\tilde{n} \times \tilde{n}}$ and $\mathbf{H} \in \mathbb{R}^{\tilde{n} \times d}$, has $\tilde{n} = \sum_i n_i + 2$ nodes: patch nodes across all scales $s_1, s_2, \ldots, s_N$ extracted from $\boldsymbol{x}_t$, as well as the two special nodes (time and context) used for conditional image generation (Figure 3 left). The augmented adjacency matrix $\tilde{\mathbf{A}} \in \mathbb{R}^{\tilde{n} \times \tilde{n}}$ is constructed using multiscale adjacency matrices across resolutions, where each scale $s_i$ has its own adjacency matrix $\mathbf{A}_{s_i}$ of grid connectivity structure. $\tilde{\mathbf{A}}$ captures both intra-resolution connections within each scale and inter-resolution connections between nodes across different scales. Specifically, nodes at finer scales are connected to the corresponding regions at coarser scales, forming a hierarchical structure. Again the two special nodes are fully connected to other patch nodes. The full block matrix representation of $\tilde{\mathbf{A}}$ is provided in Appendix A.2.

The DiG$_{\mathrm{mix}}$ block, as shown in Figure 3 (right) and outlined in Eq. 8, begins by normalizing the node features $\mathbf{H}^{(0)}$. These features are then iteratively updated through the GNN layers, utilizing the augmented adjacency matrix $\tilde{\mathbf{A}}$, for up to K hops. After the final iteration, the output features $\mathbf{H}^{(K)}$ are combined with the original $\mathbf{H}^{(0)}$ via a skip connection. The resulting features are normalized once more and passed through an MLP to yield the final output $\mathbf{H}^{[l+1]}$.

$$
\begin{aligned}
\mathbf{H}^{(k)} &= \mathrm{GNN}^{(k-1)}(\mathbf{H}^{(k-1)}, \tilde{\mathbf{A}}), \quad k = 1, \ldots, K, \quad \mathbf{H}^{(0)} = \mathrm{Norm}(\mathbf{H}^{[l]}), \\
\mathbf{H}^{[l+1]} &= \hat{\mathbf{H}} + \mathrm{MLP}(\mathrm{Norm}(\hat{\mathbf{H}})), \quad \hat{\mathbf{H}} = \mathbf{H}^{(0)} + \mathbf{H}^{(K)}.
\end{aligned}
\tag{8}
$$

### 3.3 Time and Context Nodes for Patch-Node Graphs

A notable decline in feature quality was observed during initial experimentation with DiG blocks when a fixed number of hops was applied. This degradation likely resulted from the over-smoothing effect (Oono & Suzuki, 2020; Cai & Wang, 2020; Alon & Yahav, 2020)—a common issue in GNNs. This phenomenon diminishes the model's capacity to distinguish between distinct features, thereby impairing the quality of image generation.

To evaluate the impact of over-smoothing on feature diversity, we measured feature diversity across layers using the metric $\|\mathbf{H} - \mathbf{1}\mathbf{h}^{\star T}\|$, where $\mathbf{h}^{\star} = \arg\min_{\mathbf{h}^{\star}} \|\mathbf{H} - \mathbf{1}\mathbf{h}^{\star T}\|$ (Dong et al., 2021; Han et al., 2022). As shown in Figure 4, our results indicated a pronounced reduction in feature diversity across DiG blocks, particularly when time $\mathbf{v}_t \in \mathbb{R}^{1 \times d}$ or context $\mathbf{v}_c \in \mathbb{R}^{1 \times d}$ were added $\mathbf{H}_{\mathrm{add}} = \mathbf{H} + \mathbf{v}_t +$

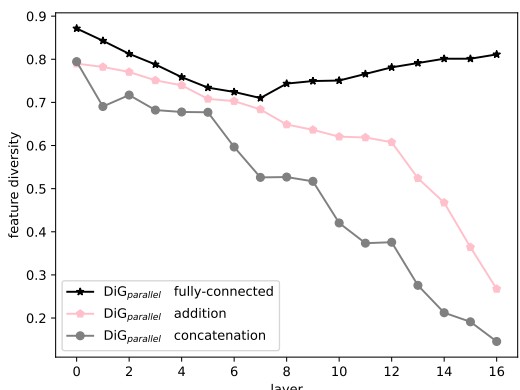

Figure 4: Feature diversity across DiG layers for DiG$_{\mathrm{parallel}}$ blocks. The plot shows how feature diversity evolves after the model converges. A similar trend is observed for the DiG$_{\mathrm{mix}}$ blocks.

$\mathbf{v}_c$; $\quad \mathbf{H}_{\mathrm{add}} \in \mathbb{R}^{n \times d}$ or concatenated $\mathbf{H}_{\mathrm{concat}} = [\mathbf{H}, \mathbf{v}_t, \mathbf{v}_c]$; $\quad \mathbf{H}_{\mathrm{concat}} \in \mathbb{R}^{n \times (3d)}$ with the node features $\mathbf{H}$ without further refinement. This approach often worsened the problem, leading to a further decline in image quality.

To address this challenge, an alternative strategy was adopted, where the time and context were treated as special nodes fully connected to all other nodes in the graph $\mathbf{H}_{\mathrm{fully\text{-}connected}} = [\mathbf{H}; \mathbf{v}_t; \mathbf{v}_c]$; $\quad \mathbf{H}_{\mathrm{fully\text{-}connected}} \in \mathbb{R}^{(n+2) \times d}$, rather than simply being added or concatenated. This fully connected configuration significantly enhanced feature diversity. The time node embedding and context embedding functioned similarly to virtual nodes in GNNs (Gilmer et al., 2017), designed to facilitate long-range information propagation. Acting as hubs, these special nodes enabled efficient communication across distant parts of the graph, mitigating the oversmoothing effect. By maintaining connectivity with all nodes, these special nodes ensured the preservation of critical information flow throughout the graph, ultimately sustaining feature diversity and improving the quality of image generation.

## 4 Experimental Results

We investigate two architectural variants for DiG: DiG$_{\mathrm{parallel}}$, which processes each resolution independently as separate graphs, and DiG$_{\mathrm{mix}}$, which operates on a multiresolution graph where nodes across scales are connected. One of the differences between these architectures lies in how they handle multiscale information. In DiG$_{\mathrm{parallel}}$, each resolution is treated in isolation, with a separate GNN applied at each scale, followed by a feature concatenation step to integrate information from all resolutions. On the other hand, DiG$_{\mathrm{mix}}$ allows information to flow between scales through hierarchical connections, directly linking fine- and coarse-grained representations.

For the remainder of the evaluation, we include both DiG$_{\mathrm{parallel}}$ and DiG$_{\mathrm{mix}}$ architectures in our experiments. We begin by assessing DiG's performance at multiple image resolutions, highlighting

Table 1: FID-10K comparison of unconditional image generation on CIFAR10. The table highlights the performance across various patch sizes, along with the corresponding number of parameters and FLOPs (in TFlops and GFlops) for each model.

| Model | Patch Size | #Params (M) | Flops | FID $\downarrow$ |
|-------|-----------|-------------|-------|------|
| $\text{DiG}_{parallel}$ | $\{2\}$ | 39.13 | 1.2T | 9.92 |
| $\text{DiG}_{parallel}$ | $\{2, 4\}$ | 58.05 | 1.72T | 6.88 |
| $\text{DiG}_{parallel}$ | $\{2, 4, 8\}$ | 65.14 | 1.85T | 6.92 |
| $\text{DiG}_{mix}$ | $\{2\}$ | 37.48 | 1.53T | 12.72 |
| $\text{DiG}_{mix}$ | $\{2, 4\}$ | 37.59 | 1.61T | 11.23 |
| $\text{DiG}_{mix}$ | $\{2, 4, 8\}$ | 37.99 | 1.63T | 10.44 |
| U-ViT | 2 | 44.26 | 1.45T | **6.07** |
| U-ViT | 4 | 44.19 | 0.37T | 16.58 |
| U-ViT | 8 | 44.32 | 96.41G | 41.72 |
| DiT | 2 | 57.78 | 1.24T | 8.23 |
| DiT | 4 | 57.72 | 0.31T | 18.23 |
| DiT | 8 | 57.84 | 80.35G | 37.23 |

its capacity to capture information at different scales. This is followed by a comprehensive evaluation of DiG on both unconditional and class-conditional image generation tasks. A summary of the main experimental setup is provided below, with further details, including network architectures and sampling hyperparameters, available in Appendix C.

## 4.1 EXPERIMENTAL SETUP

Both conditional and unconditional image generation tasks were evaluated on the CIFAR10 dataset Krizhevsky et al. (2009), which consists of 50K images at a resolution of 32x32 pixels across 10 distinct classes. Additionally, the SVHN dataset (Netzer et al., 2011), containing 73K images of resolution 32x32, was used for unconditional image generation. Experiments were also conducted on the ImageNet dataset (Deng et al., 2009), with images at a 64x64 resolution, utilizing $1,287,167$ images across 1,000 labels.

The AdamW optimizer (Loshchilov et al., 2017) was utilized with a weight decay of $0.3$ and a learning rate of 2e-4 and 3e-4. All models were trained for an equal number of iterations while maintaining a relatively consistent number of parameters across experiments. For image sampling, the Euler-Maruyama SDE (Song et al., 2021b) or DPM-Solver (Lu et al., 2022) was used, with the same number of steps applied to ensure fair comparison. Both DiG architectures are compared against U-ViT[1] (Bao et al., 2023) and DiT[2,3] (Peebles & Xie, 2023).

## 4.2 IMPACT OF MULTI-SCALE RESOLUTIONS

We begin by assessing the impact of incorporating multi-scale image resolutions in graph-based diffusion models. It is well-established that diffusion transformers tend to experience a decline in image quality as patch sizes increase (Bao et al., 2023). Table 1 compares $\text{DiG}_{parallel}$ and $\text{DiG}_{mix}$ across various patch sizes. See Appendix B for a comparison of different GNN layers on CIFAR10 and Appendix D for additional results.

Unlike diffusion transformers, which typically exhibit worse FID scores with larger patches, DiG architectures maintain competitive FID scores across different patch sizes, albeit with higher parameter and FLOP costs. Notably, $\text{DiG}_{parallel}$ achieves a strong FID-10K score at patch sizes $\{2, 4\}$ with 6.88, comparable to U-ViT at 6.07. Representing larger patches allows the model to capture global information, crucial for modeling broader spatial dependencies in image generation.

---

[1] https://github.com/baofff/U-ViT/tree/main

[2] https://github.com/facebookresearch/DiT

[3] A fixed covariance was used in our experiments, while DiT learns both mean and covariance. The decoder block was modified for a fair comparison, which resulted in some performance degradation.

Table 2: Models across different datasets. FID-50K scores, FLOPs, and parameter counts are presented for each model. Complete model specifications and further details can be found in Appendix C.

| Model | #Params | Flops | FID ↓ | Model | #Params | Flops | FID ↓ |
|---|---|---|---|---|---|---|---|
| | **Unconditional CIFAR-10** | | | | **Conditional CIFAR-10** | | |
| U-ViT/2 | 44.26M | 1.45T | 6.05 | U-ViT/2 | 44.26M | 1.46T | **2.95** |
| DiT/2 | 57.78M | 1.24T | **5.76** | DiT/2 | 57.78M | 1.24T | 4.69 |
| DiG$_{parallel}$/$\{2,4\}$ | 58.05M | 1.72T | 6.16 | DiG$_{parallel}$/$\{2,4\}$ | 58.06M | 1.73T | 4.56 |
| DiG$_{mix}$/$\{2,4\}$ | 37.59M | 1.61T | 9.47 | DiG$_{mix}$/$\{2,4\}$ | 40.6M | 1.62T | 6.47 |

| Model | #Params | Flops | FID ↓ | Model | #Params | Flops | FID ↓ |
|---|---|---|---|---|---|---|---|
| | **SVHN** | | | | **ImageNet 64x64** | | |
| U-ViT/2 | 28.68M | 0.94T | **2.47** | U-ViT/4 | 130.94M | 4.29T | **9.17** |
| DiT/2 | 37.04M | 0.79T | 3.93 | DiT/4 | 173.06M | 3.72T | 17.36 |
| DiG$_{parallel}$/$\{2,4\}$ | 37.57M | 1.11T | 2.96 | DiG$_{parallel}$/$\{4,8\}$ | 125.88M | 3.75T | 17.00 |
| DiG$_{mix}$/$\{2,4,8\}$ | 24.74M | 1.05T | 3.22 | DiG$_{mix}$/$\{4,8\}$ | 111.05M | 4.53T | 22.00 |

Empirical results demonstrated that DiG$_{parallel}$ consistently outperformed DiG$_{mix}$ in terms of FID score for image quality. This suggests that processing each resolution separately before combining them enables more detailed and distinct feature extraction at each scale. However, this performance improvement comes at the expense of increased computational cost.

In DiG$_{parallel}$, after processing each resolution, the features from all scales are concatenated across multiple layers $\hat{\mathbf{H}}$ (Eq. 5). This concatenation step requires the model to manage and process a high-dimensional feature space, particularly when multiple resolutions are used, as the dimensionality of the concatenated features scales with $d \times |\mathcal{P}|$. The concatenated features are subsequently passed through an MLP for transformation, further increasing computational overhead. The complexity grows linearly with the number of resolutions, making DiG$_{parallel}$ computationally more expensive, especially as the number of layers $L$ increases. To address this computational burden, we experimented with summing the features from different resolutions rather than concatenating them. While this alternative approach would reduce the dimensionality and computational load, it led to a noticeable degradation in image quality (also see Section 3.3).

### 4.3 IMAGE GENERATION RESULTS

Table 2 presents the results for image generation, where DiG$_{parallel}$ demonstrates competitive performance across multiple datasets, particularly in balancing image quality and capturing multiscale features. More details on the model architecture and training settings are provided in Appendix C.

Although diffusion transformer-based models like U-ViT and DiT generally perform well in terms of FID-50K, DiG$_{parallel}$ maintains robust performance, especially at patch sizes $\{2,4\}$, while offering additional flexibility through its multiscale graph-based architecture. This highlights the effectiveness of the DiG approach in handling different scales while retaining strong generative capabilities.

This demonstrates the potential of a graph-based architecture to efficiently capture multiscale features while maintaining competitive performance in image generation tasks. DiG$_{mix}$ is parameter efficient and offers a more computationally lightweight alternative to DiG$_{parallel}$, albeit with a slight performance trade-off. Examples of generated images for each dataset are shown in Figure 5.

## 5 CONCLUSIONS

In conclusion, this paper presents a novel exploration of graph-based architectures for image generation within diffusion models, challenging the dominance of transformer-based approaches. We propose DiG, a model that introduces two variants: DiG$_{parallel}$, which processes each resolution independently, and DiG$_{mix}$, which captures multiscale information by connecting nodes across different scales. Our experiments demonstrate that GNNs can effectively replace transformers for image gen-

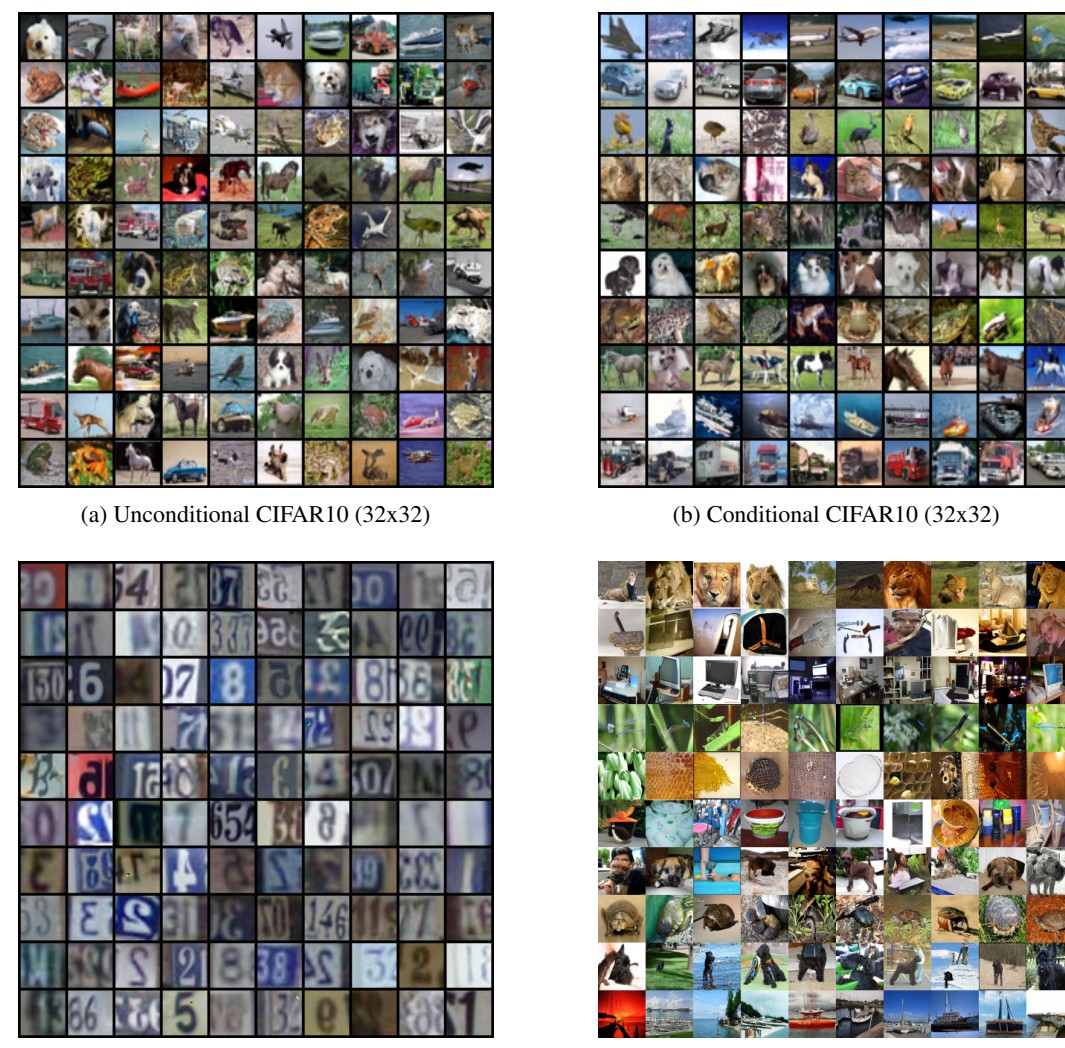

(a) Unconditional CIFAR10 (32x32)    (b) Conditional CIFAR10 (32x32)

(c) SVHN (32x32)    (d) Imagenet (64x64)

Figure 5: Sample images generated across multiple datasets.

eration, providing a competitive alternative while maintaining high-quality outputs across multiple datasets. The promising results of our DiG models suggest that graph-based methods offer significant potential for future work in visual generative modeling. We hope this study will inspire the research community to further explore GNN-based architectures for image generation and scaling experiments to larger models.

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

# A   ADJACENCY MATRIX REPRESENTATION

## A.1   BLOCK MATRIX FOR DIG$_{\text{PARALLEL}}$

In DiG$_{\text{parallel}}$, the augmented adjacency matrix $\tilde{\mathbf{A}}_{s_1}$ includes the grid-structured patch adjacency matrix $A_{s_1}$, along with time and context nodes fully connected to all patches.

$$
\tilde{\mathbf{A}}_{s_1} = \begin{bmatrix} \mathbf{A}_{s_1} & \mathbf{a}_t & \mathbf{a}_c \\ \mathbf{a}_t^{\top} & 0 & 0 \\ \mathbf{a}_c^{\top} & 0 & 0 \end{bmatrix}
$$

where $\mathbf{A}_{s_1} \in \mathbb{R}^{n_1 \times n_1}$ represents the grid connections between patches. $\mathbf{a}_t \in \mathbb{R}^{n_1}$ is the connection of the time node with all patch nodes. $\mathbf{a}_c \in \mathbb{R}^{n_1}$ is the connection of the context node with all patch nodes.

### A.1.1   POOLING MASK $\mathbf{C}_{s_i}$

For any coarser scale $s_i$, the pooling matrix $\mathbf{C}_{s_i}$ of size $\tilde{n}_i \times \tilde{n}_1$ can be written as:

$$
\mathbf{C}s_i = \begin{bmatrix} c_{11} & c_{12} & \cdots & c_{1n_1} & 0 & 0 \\ c_{21} & c_{22} & \cdots & c_{2n_1} & 0 & 0 \\ \vdots & \vdots & \ddots & \vdots & 0 & 0 \\ c_{n_i1} & c_{n_i2} & \cdots & c_{n_in_1} & 0 & 0 \\ 0 & 0 & \cdots & 0 & 1 & 0 \\ 0 & 0 & \cdots & 0 & 0 & 1 \end{bmatrix}
$$

where the submatrix $c_{ij}$ corresponds to pooling the patch nodes from the finer scale $s_1$ into the coarser scale $s_i$, reducing the number of nodes. The last two rows preserve the time node and the context node.

## A.2   BLOCK MATRIX FOR DIG$_{\text{MIX}}$

In DiG$_{\text{mix}}$, the adjacency matrix $\tilde{\mathbf{A}}$ includes connections within each resolution (intra-resolution) and between resolutions (inter-resolution), along with time and context nodes.

$$
\tilde{\mathbf{A}} = \begin{bmatrix} \mathbf{A}_{s_1} & \mathbf{A}_{s_1,s_2} & \cdots & \mathbf{A}_{s_1,s_N} & \mathbf{a}_t & \mathbf{a}_c \\ \mathbf{A}_{s_1,s_2}^{\top} & \mathbf{A}_{s_2} & \cdots & \mathbf{A}_{s_2,s_N} & \mathbf{a}_t & \mathbf{a}_c \\ \vdots & \vdots & \ddots & \vdots & \vdots & \vdots \\ \mathbf{A}_{s_1,s_N}^{\top} & \mathbf{A}_{s_2,s_N}^{\top} & \cdots & \mathbf{A}_{s_N} & \mathbf{a}_t & \mathbf{a}_c \\ \mathbf{a}_t^{\top} & \mathbf{a}_t^{\top} & \cdots & \mathbf{a}_t^{\top} & 0 & 0 \\ \mathbf{a}_c^{\top} & \mathbf{a}_c^{\top} & \cdots & \mathbf{a}_c^{\top} & 0 & 0 \end{bmatrix}
$$

where $\mathbf{A}_{s_i} \in \mathbb{R}^{n_i \times n_i}$ represents intra-resolution connections for scale $s_i$. $\mathbf{A}_{s_i,s_j} \in \mathbb{R}^{n_i \times n_j}$ represents inter-resolution connections between scale $s_i$ and $s_j$. $\mathbf{a}_t \in \mathbb{R}^{\sum_i n_i}$ is the connection of the time node with all patch nodes. $\mathbf{a}_c \in \mathbb{R}^{\sum_i n_i}$ is the connection of the context node with all patch nodes.

# B   IMPACT OF TYPE OF GNN LAYER

The impact of different GNN layer types was evaluated in this study. Specifically, GCN (Kipf & Welling, 2016), GraphSAGE (Hamilton et al., 2017), and GIN (Xu et al., 2018) were tested, as shown in Table 3. CIFAR10 was used in the unconditioned setting to compare the FID-10K scores across various GNN layers. In all cases, two hops $K = 2$ were selected, as additional hops did not yield performance improvements. GraphSage demonstrated better performance and was selected for the remaining experiments.

Table 3: FID-10K scores for different GNN layers on unconditional CIFAR10 with patch sizes $\mathcal{P} = \{2, 4\}$

|  | GCN | GIN | GraphSAGE |
|---|---|---|---|
| $\text{DiG}_{parallel}$ | 8.33 | 9.23 | 6.88 |
| $\text{DiG}_{mix}$ | 9.93 | 11.99 | 8.23 |

Table 4: Experimental setup for DiG, DiT, and U-ViT architectures for results presented in the main paper.

| Dataset | CIFAR10 | SVHN | ImageNet 64×64 |
|---|---|---|---|
| | DiG | | |
| K-hops | 2 | 2 | 1 |
| GNN Convolution | GraphSAGE | GraphSAGE | GraphSAGE |
| | U-ViT | | |
| #Heads | 8 | 8 | 12 |
| | DiT | | |
| #Heads | 8 | 8 | 12 |
| Learn Sigma | False | False | False |
| Embedding dimension | 512 | 448 | 768 |
| Depth | 12 | 10 | 16 |
| Batch size | 128 | 256 | 512 |
| Training iterations | 500K | 400K | 300K |
| Warm-up steps | 2.5K | 2.5K | 5K |
| Optimizer | AdamW | AdamW | AdamW |
| Learning rate | 2e-4 | 2e-4 | 3e-4 |
| Weight decay | 0.03 | 0.03 | 0.03 |
| Betas | (0.99, 0.999) | (0.99, 0.99) | (0.99, 0.99) |
| Noise schedule | VP | VP | VP |
| Sampler | Euler-Maruyama | Euler-Maruyama | DPM-Solver |
| Sampling steps | 1K | 1K | 50 |

## C  NETWORKS DETAILS

The DiG, DiT, and U-ViT architectures were trained as part of the experimental setup for the results presented in the main paper. Table 4 summarizes the experimental setup for $\text{DiG}_{\text{parallel}}$, $\text{DiG}_{\text{mix}}$, DiT and U-ViT across different datasets. "VP" represents the continuous-time variance preserving noise schedule (Song et al., 2021b).

Teh Euler-Maruyama (Song et al., 2021b) sampler was used for CIFAR10 and SVHN, while DPM-Solver (Lu et al., 2022) was employed for ImageNet 64×64.

Also see Appendix B for a comparison of different GNN layers on CIFAR10.

## D  ADDITIONAL COMPARISON RESULTS

Figure 6 shows the loss and FID curves for the best results of unconditional CIFAR10 shown in Table 1.

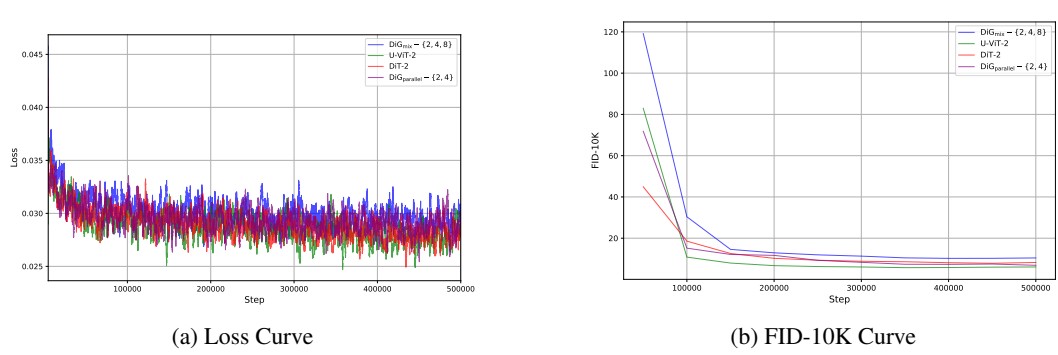

(a) Loss Curve

(b) FID-10K Curve

Figure 6: Loss and FID-10K curves for unconditional CIFAR10.

