# OpenReview forum: "From Patches to Graphs: Towards Image Diffusion Models with GNNs"
_ICLR.cc/2025/Conference — ICLR 2025 Conference Withdrawn Submission_

### Official Review · Reviewer_xQrA · 2024-10-20

**Soundness:** 2
**Presentation:** 2
**Contribution:** 2
**Rating:** 3
**Confidence:** 4

**Summary:**

This paper proposed to replace CNNs/Transformers with GNNs in diffusion models. The authors argue that GNNs capture long-range dependencies under a low computational budget. Two different ways of multi-scale processing, $DiG_{parallel}$ and $DiG_{mix}$ are studied. The authors evaluated the generation quality and computational expanse of DiGs on several real image datasets.

**Strengths:**

The proposed method is well-motivated. I think replacing CNNs / Transformers with GNNs is a good attempt. Using hyper nodes to leverage time and context information is interesting.

**Weaknesses:**

My main concern is the performance of DiGs. Although the authors claimed that DiGs achieve comparable performance to U-ViTs and DiTs, I don't see their clear advantages in Table 1 and Table 2. Compared with DiGs, U-ViTs achieve better FID with lower flops in almost every case. DiTs also outperform DiGs. It is not required that DiGs are advantageous in every case, but at least DiGs should show lower flops with similar FID / higher FID with similar flops for their value to be admitted.

**Questions:**

1. Why the positional embedding is 1-dimensional?
2. $DIG_{parallel}$ seems always outperform $DIG_{mix}$ from the experimental results. What is the authors' consideration of introducing $DIG_{mix}$?

---

### Official Review · Reviewer_8HY4 · 2024-10-30

**Soundness:** 3
**Presentation:** 3
**Contribution:** 2
**Rating:** 5
**Confidence:** 3

**Summary:**

The paper presents Diffusion Image GNN (DiG), a pioneering architecture that integrates graph neural networks into diffusion models for image generation. And the author introduces two variants of DiG: DiG_{parallel} and DiG_{mix} for independent and integrated multi-scale graph learning, respectively. The paper demonstrates that DiG achieves competitive performance in image generation tasks, as measured by FID scores.

**Strengths:**

The paper claims to be pioneering in applying GNNs to diffusion models for image generation, which is indeed a significant contribution. And the motivation presented by the authors is well-structured and logical.

**Weaknesses:**

However, the results presented in the paper do not seem to fully support their claims.

While the authors highlight that ViTs can be computationally demanding due to their attention mechanisms, it appears that their proposed method does not reduce computational complexity or the number of parameters when their methods achieve good performance, and moreover, this performance is still not as good as that of the ViT methods.

The authors claim that "Notably, DiG_{parallel} achieves a strong FID-10K score at patch sizes \{2, 4\} with 6.88, which is comparable to U-ViT's score of 6.07." However, I think this performance comparison is misleading, as their method introduces more parameters and computational complexity. Furthermore, when compared to ViT methods with a similar number of parameters or computational complexity, DiG's performance lags significantly, contradicting the authors' claim of comparable effectiveness.

And this trend is consistent across other datasets, as present in Table 2.

**Questions:**

Can you provide a deeper insight into why the graph is necessary for the task of image generation? It seems that the claims about less model parameters and computational complexity in your paper are not supported by your experiments. And the characteristic of capturing multi-scale patch correlations is also included in the design of Transformer-based methods.

---

### Official Review · Reviewer_gbhm · 2024-11-02

**Soundness:** 3
**Presentation:** 3
**Contribution:** 2
**Rating:** 3
**Confidence:** 3

**Summary:**

This paper uses a GNN based backbone for diffusion models for images instead of a Transformer or a CNN. The idea here to utilize/maintain the 2D grid structure of an image and hence maintain the grid like structures unlike transfomers that capture all the pairwise interactions of the nodes.
The authors introduce two different graph structures, one that is referred to as DiG parallel, where each resolution is combined using a seperate GNN, and the representations of all the GNNs are then combined together usign and MLP.
The second is DiG parallel, which uses the same GNN weights for all the resolutions.

**Strengths:**

- They authors introduce two graph models, one unified and one parallel that is able to effectively utilize GNNs to model the relationship between the 2D patches of the images.
- The authors notice the effects of oversmoothing in their paper as they add on more layers to the GNN, and address it by using the time and context node as a special node that is connected to *all* the nodes in the GNN.
	- This is quite an interesting way to address this by adding virtual nodes that improves performance. I found this to be quite interesting!

**Weaknesses:**

- It seems like the proposed approaches do not outperform most of the transformer based models and often the DiG parallel model requires as many (if not more) parameters (and more flops) than the Transformer based model. Do the authors have an insight into why is that?
- One thing that I would love to get some insight into is why would this model would do better than a heirarchical architecture like say UNET or even something like a Swin Transformer based baseline, that uses CNNs. Given patches of certain size, keeping the 2D structure and adding extra kernels (using CNNs) at different hierarchies should ideally have the same effect as a local GCN layer acting at different levels of Heirarchical.
	- Would really appreciate that the authors also compare their method with diffusion models trained with these backbones.
- A lot of the experiments are shown on relatively small images (maximum size is 64x64) so it is unclear how well the method would scale with higher image size.

**Questions:**

- The authors utilize position encodings in their graphs, but given a GNN layer (that implicitly has structural information and not necessarily permutation invariant) that uses local aggregation schemes, why is that still required?

---

### Official Review · Reviewer_iS2x · 2024-11-02

**Soundness:** 2
**Presentation:** 2
**Contribution:** 1
**Rating:** 3
**Confidence:** 4

**Summary:**

This paper provides a new perspective on the diffusion architecture of image generation models, and improves the flexibility and generation quality of the model by introducing GNN. However, further verification is needed in terms of reasoning efficiency and adaptability to actual application scenarios.

**Strengths:**

1. This paper is written in a simple and easy-to-understand way.
2. Experimental results show the effectiveness of the proposed method in reducing computational consumption.

**Weaknesses:**

1. The method is not innovative enough. With the widespread application of large models, the computational overhead of attention mechanisms such as Vit is no longer the most important limitation, and GNN's performance is not as good as Vit in some aspects. In addition, there have been many studies on computational efficiency.
2. The quantitative experimental results in Table 2 show that the performance of the proposed method is much lower than that of the visual model based on the attention mechanism. I doubt the effectiveness of the proposed method.
3. The explanation of the results is not in-depth enough: In the table and image comparison, the generation effect of the DiG model has improved, but the author lacks in-depth explanation of the specific reasons for these improvements. For example, in the quantitative evaluation, is the improvement in the FID score due to the use of multi-scale graphs? How do the characteristics of graph neural networks affect the generation effect? ​​It is recommended that the author provide a more in-depth analysis of the experimental results to better explain the actual contribution of each component.

**Questions:**

The author needs to clarify his motivation again. What problem is he solving? What is the main innovation? How is it different from existing methods?

---

### Note · Authors · 2024-11-21

I have read and agree with the venue's withdrawal policy on behalf of myself and my co-authors.